# Identification of a large anion channel required for digestive vacuole acidification and amino acid export in *Plasmodium falciparum*

Gagandeep S. Saggu[1¤], Jinfeng Shao[1], Mansoor A. Siddiqui[1], Maria Traver[2], Tatiane Macedo-Silva[1], Joseph Brzostowski[2], Sanjay A. Desai[1]*

1 Laboratory of Malaria and Vector Research, National Institute of Allergy and Infectious Diseases, National Institutes of Health, Rockville, Maryland, United States of America, 2 Twinbrook Imaging Facility, Laboratory of Immunogenetics, National Institute of Allergy and Infectious Diseases, National Institutes of Health, Rockville, Maryland, United States of America

¤ Current address: College of Pharmacy, The Ohio State University, Columbus, Ohio, United States of America

* sdesai@niaid.nih.gov

**Academic editor:** Tania F. de Koning-Ward, Deakin University—Geelong Waurn Ponds Campus, AUSTRALIA

## Abstract

Malaria parasites survive in human erythrocytes by importing and digesting hemoglobin within a specialized organelle, the digestive vacuole (DV). Although chloroquine and other antimalarials act within the DV, the routes used by drugs, ions, and amino acids to cross the DV membrane remain poorly understood. Here, we used single DV patch-clamp to identify a novel large conductance anion channel as the primary conductive pathway on this organelle in *Plasmodium falciparum*, the most virulent human pathogen. This Big Vacuolar Anion Channel (BVAC) is primarily open at the DV resting membrane potential and undergoes complex voltage-dependent gating. Ion substitution experiments implicate promiscuous anion flux with Cl⁻ being the primary charged substrate under physiological conditions. Conductance and gating are unaffected by antimalarials targeting essential DV activities and are conserved on parasites with divergent drug susceptibility profiles, implicating an unexploited antimalarial target. A conditional knockdown strategy excluded links to PfCRT and PfMDR1, two drug-resistance transporters with poorly defined transport activities. We propose that BVAC functions to maintain electroneutrality during H⁺ uptake, allowing DV acidification and efficient hemoglobin digestion. The channel also facilitates amino acid salvage, providing essential building blocks for parasite growth. Direct transport measurements at the DV membrane provide foundational insights into vacuolar physiology, should help clarify antimalarial action and drug resistance, and will guide therapy development against the parasite's metabolic powerhouse.

## Introduction

Parasites of the genus *Plasmodium* invade and replicate within host erythrocytes to evade immune responses and acquire hemoglobin as an amino acid source. In the

**Data availability statement:** The data that support this study's findings are included in the manuscript and its Supporting information. The computer code used for the analyses of single channel recordings in DIAdem 2015 (National Instruments) is available through an open repository (https://www.ni.com/en/support/downloads/dataplugins/download.axon-instruments-dataplugin-for-axon-abf.html#372093). Additional code tailored specifically for the channel described here can be found at https://doi.org/10.5281/zenodo.15305314.

**Funding:** This work was supported by the Division of Intramural Research, National Institute of Allergy and Infectious Diseases, National Institutes of Health (ZIA# AI000882 (2025) to S.A.D). M.A.S. was supported by an NIAID Malaria Research Program Fellowship. The funders had no role in study design, data collection and analysis, decision to publish, or preparation of the manuscript.

**Competing interests:** The authors have declared that no competing interests exist.

**Abbreviations:** BVAC, Big Vacuolar Anion Channel; DAPI, 4′,6-diamidino-2-phenylindole; DDD, DHFR degradation domain; DV, digestive vacuole; HRP, horse radish peroxidase; NeoR, neomycin resistance gene; TMP, trimethoprim.

virulent human parasite, *Plasmodium falciparum*, hemoglobin is endocytosed and delivered to the digestive vacuole (DV), a specialized acidic organelle [1]. There, it is digested by aspartic and cysteine proteases [2,3], providing amino acids for parasite protein synthesis. Heme liberated by this process is rapidly detoxified within the DV through biomineralization into hemozoin [4]. The DV may also regulate infected erythrocyte water balance and metabolize parasite lipids [5].

Consistent with its central role in parasite metabolism, the DV is the target of multiple antimalarial drugs that act either within this organelle or against its membrane transport activities. Chloroquine, once the mainstay of malaria prophylaxis and treatment, interferes with heme detoxification [6]. This inexpensive and safe drug has been sidelined by acquired parasite resistance attributed to mutations in PfCRT and PfMDR1, two putative transporters at the DV membrane [7]. Resistance to diverse antimalarials including mefloquine, piperaquine, lumefantrine, and artemisinins has also been linked to mutations or gene amplification of PfMDR1 and other putative transporters proposed for the DV membrane [8–10].

Although the parasite depends on DV trafficking of H⁺, amino acids, antimalarial drugs, and ions (Fig 1A), our understanding of DV membrane transport properties remains rudimentary. Macroscopic tracer flux studies using either crude DV preparations or heterologous expression of candidate DV transporters have yielded conflicting proposals for membrane transport activities and antimalarial resistance mechanisms [11–16]. To address this deficit, we now report direct measurement of transport at this heretofore inaccessible membrane. We used patch-clamp of rapidly isolated, viable DVs to identify a constitutively active, large-conductance anion channel as the primary ion channel on this membrane. Our findings implicate two essential roles in maintaining electroneutrality during acidification and facilitating amino acid efflux. Direct transport measurements on single DVs provide a framework for understanding the unique physiology of this important drug target.

## Results

### A conserved, large-conductance channel on the DV

We modified existing parasite fractionation protocols to obtain rapid harvest of intact, metabolically active DVs. We confirmed faithful recovery using visual examination and TEM (Fig 1A, 1B). We used Dd2^attB/CRT-GFP, an engineered parasite with the DV-resident membrane PfCRT protein carrying a C-terminal GFP tag, to establish unambiguous identification of membrane-bound DVs (S1 Fig). E(1)GFP-tagged plasmepsin II, a soluble protein targeted to the DV lumen, was retained in harvested DVs, suggesting preserved membrane integrity throughout our harvest protocol (S2 Fig). To study membrane transport on this small organelle (~1 μm diameter), we performed patch-clamp with small-tipped patch pipettes. To facilitate filling of narrow-bore pipettes, reduce access resistance and enable high-resistance seals on the DV membrane, we implemented microscopic heating with simultaneous application of positive internal pressure to expand the pipette shank (Fig 1C), a process similar to glassblowing. The DV remained metabolically active during patch-clamp, as indicated by ongoing hemozoin tumbling (S1 Video). As previously reported [4,17], active

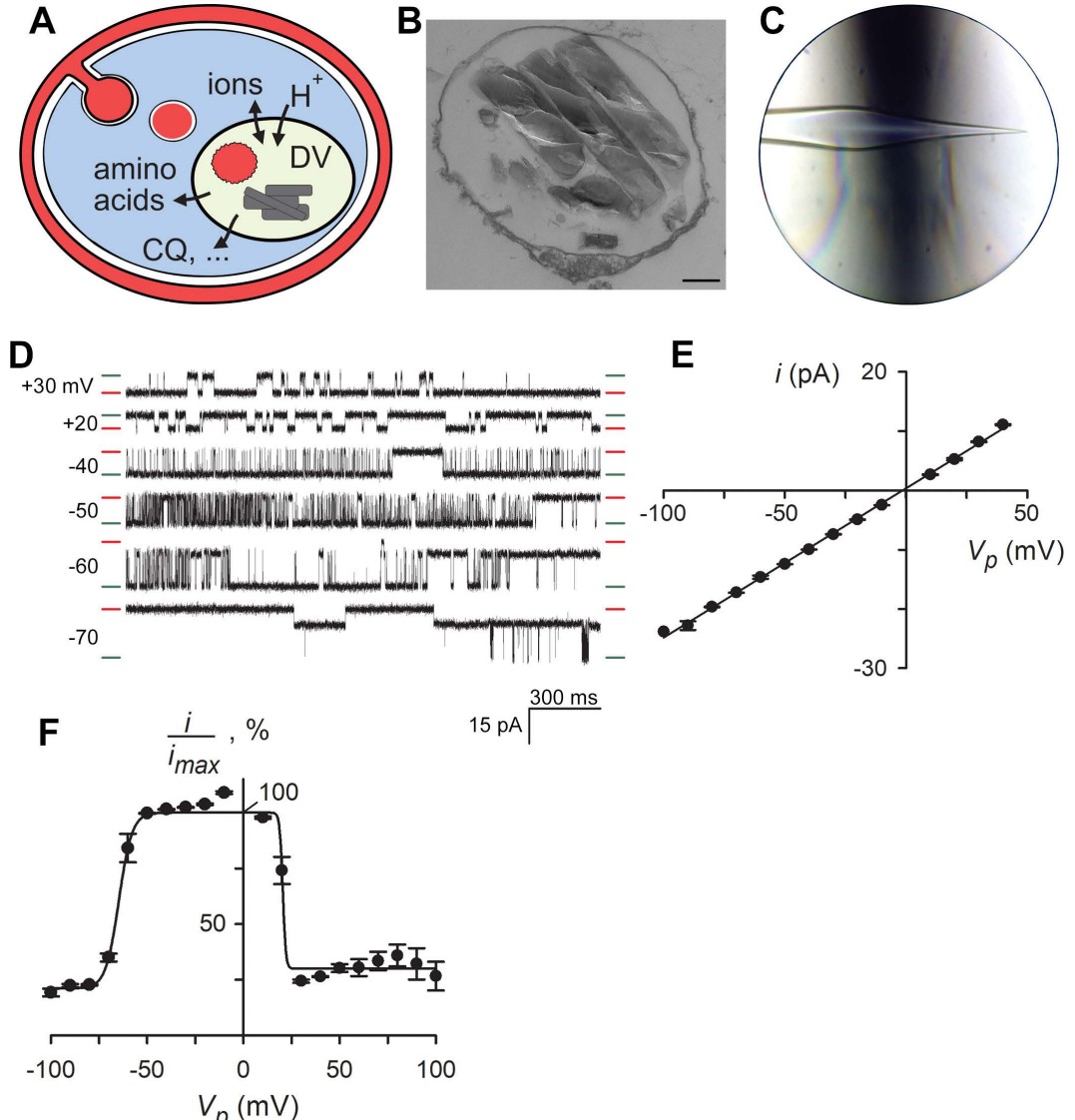

**Fig 1. A large conductance ion channel on the DV membrane. (A)** Schematic of a *Plasmodium*-infected erythrocyte showing endocytosis of erythrocyte cytosol, hemoglobin digestion in the DV, and multiple transport activities at the DV membrane. **(B)** Transmission electron micrograph of harvested DV showing intact membrane and luminal hemozoin crystals. Scale bar, 200 nm. **(C)** Patch-clamp pipette after expansion of the shank proximal to the pipette tip via positive internal pressure while heating. The heating filament is visible as a dark haze behind the pipette. The sharp, narrow bore tip is preserved, permitting DV capture and patch-clamp. **(D)** Recordings from a DV membrane patch with one functional ion channel. Imposed pipette potentials are indicated to the left of each trace; closed and open channel current levels are indicated with red and green dashes, respectively. Stable intermediate current levels reflect subconductance states. Buffer A with WOS additive in bath and pipette, S2 Table. **(E)** Current-voltage (*i*-V) relationship for the identified channel. Mean ± S.E.M. measured single channel current amplitudes, *i*, for transitions between fully closed and open levels at each imposed pipette potential ($V_p$). **(F)** Open probability at each $V_p$, calculated as the integrated current normalized to the maximum current associated with a fully open channel, $i_{max}$. Solid line, best fit to Eq. 2. The underlying data can be found at https://doi.org/10.5281/zenodo.15305314.

hemozoin movement was seen in intact infected cells and freshly harvested DVs (S2 and S3 Videos), but was lost ~1 h after isolation (S4 Video). Notably, because hemozoin tumbling occurs by an unknown mechanism, the link between fresh DV harvest and sustained metabolic activity in our experiments remains circumstantial.

Using a buffered solution with 70 mM $Na^+$, 70 mM $K^+$, and 140 mM $Cl^-$ as the primary charge carriers, we identified a large conductance channel on the DV membrane (Fig 1D). The combination of high-resistance seals on the DV membrane and the channel's large conductance permitted confident identification of open and closed channel levels (S3A Fig). We named this channel Big Vacuolar Anion Channel (BVAC) based on the studies described below. Currents at a range of voltages yielded a linear current–voltage relationship with a slope conductance of $300 \pm 22$ pS ($n = 9$ molecules; Fig 1E), with molecule-to-molecule variability attributed to complex and long-lasting subconductance states. Gating was steeply voltage-dependent with channels open nearly 100% of the time at the resting DV membrane potential and an asymmetric bell-shaped open probability profile (Fig 1F). Voltage-dependent gating was also readily apparent in single-channel recordings, where modest changes in the electric field had marked effects on single-channel behavior (Fig 1D).

As frequently used in parasite fractionation studies, our DV harvest procedure entails infected cell lysis with saponin, a glycoside detergent that might produce channel-like pores in membranes. To exclude possible artifacts, we harvested DVs without the use of detergents and identified BVAC activity with indistinguishable properties (S3B Fig). Our initial patch-clamp surveys included glucose, ATP, GTP, and phosphocreatine in the recording solutions to allow detection of energy-dependent channels. BVAC gating and conductance were unaffected by removal of these energy sources (S3C Fig), excluding their requirement in gating or transport. These changes also had no effect on the channel's complex gating or multiple subconductance levels (S3B–S3D Fig). Adding to this complex behavior, some channels appear to form functional dimers on the DV membrane (S3E Fig). Examination of consecutively recorded traces at an imposed pipette potential ($V_p$) of −60 mV revealed that this molecule exhibited periods of fast flickering transitions between dominant states, extended dwell in either open or closed states, and intermittent changes in gating (S4 Fig), establishing fidelity and stability of recordings on DV membranes [18].

Over the past 70 years, chloroquine, mefloquine, and other antimalarials that interfere with DV function have produced strong selective pressure, leading to mutations and copy number variations for genes encoding transport proteins on the DV [7,19]. We therefore performed DV patch-clamp with Dd2, 3D7, and HB3, clonal lines with distinct antimalarial drug resistance patterns and originating from patients in Indochina, South Africa (collected in the Netherlands), and Honduras respectively [20]. Despite differing antimalarial susceptibilities and geographical origins, these clones exhibit indistinguishable BVAC conductance, gating, and voltage dependence (S5 Fig), indicating strict conservation of the channel and its biophysical properties.

## Broad selectivity for anions including the amino acid glutamate⁻

To investigate potential biological roles for this newly identified DV channel, we conducted solute selectivity studies. Using gigaseals formed with pipette and bath solutions having 100 mM $CaCl_2$ and 50 mM KCl respectively, we created inward gradients for $Ca^{++}$ and $Cl^-$ and an outward $K^+$ gradient at the DV membrane. Consistent with the formation of these gradients and a selective channel, we detected reproducible channel activity in the absence of an imposed membrane potential (Fig 2A). The reversal potential, $E_{rev}$ of + 26 mV, approximated the $Cl^-$ Nernst equilibrium potential, $E_{Cl-}$ of +31 mV, but a cation channel with a preference for $K^+$ over $Ca^{++}$ cannot not be excluded by this experiment alone. To distinguish between these possibilities, we next replaced the primary charge carriers in the pipette solution with uncharged sorbitol (Fig 2B) to isolate ion efflux from the DV. Under these conditions, channel activity was again detected without imposed membrane potential, but with positive associated currents; a negative shift in $E_{rev}$ to ~ −20 mV established greater $Cl^-$ permeability ($P_{Cl}/P_{cation} \geq 2.6$). Limitations associated with small organelle patch-clamp [21] and our unsuccessful attempts to obtain reproducible whole-organelle recordings prevented formal exclusion of BVAC cation permeability.

Additional bi-ionic experiments revealed high permeability to other anions, with $Br^-$, $SCN^-$, and the divalent anions $SO_4^{-2}$ and $HPO_4^{-2}$ all exhibiting permeabilities comparable to $Cl^-$ based on $E_{rev}$ measurements. Because $E_{rev}$ is established at a voltage where infinitesimally small fluxes of the two ions are equal and opposite, this approach is a thermodynamic measurement not affected by the sizes or mobilities of the two ions. With the amino acid glutamate⁻ in the pipette, smaller

PLOS Biology

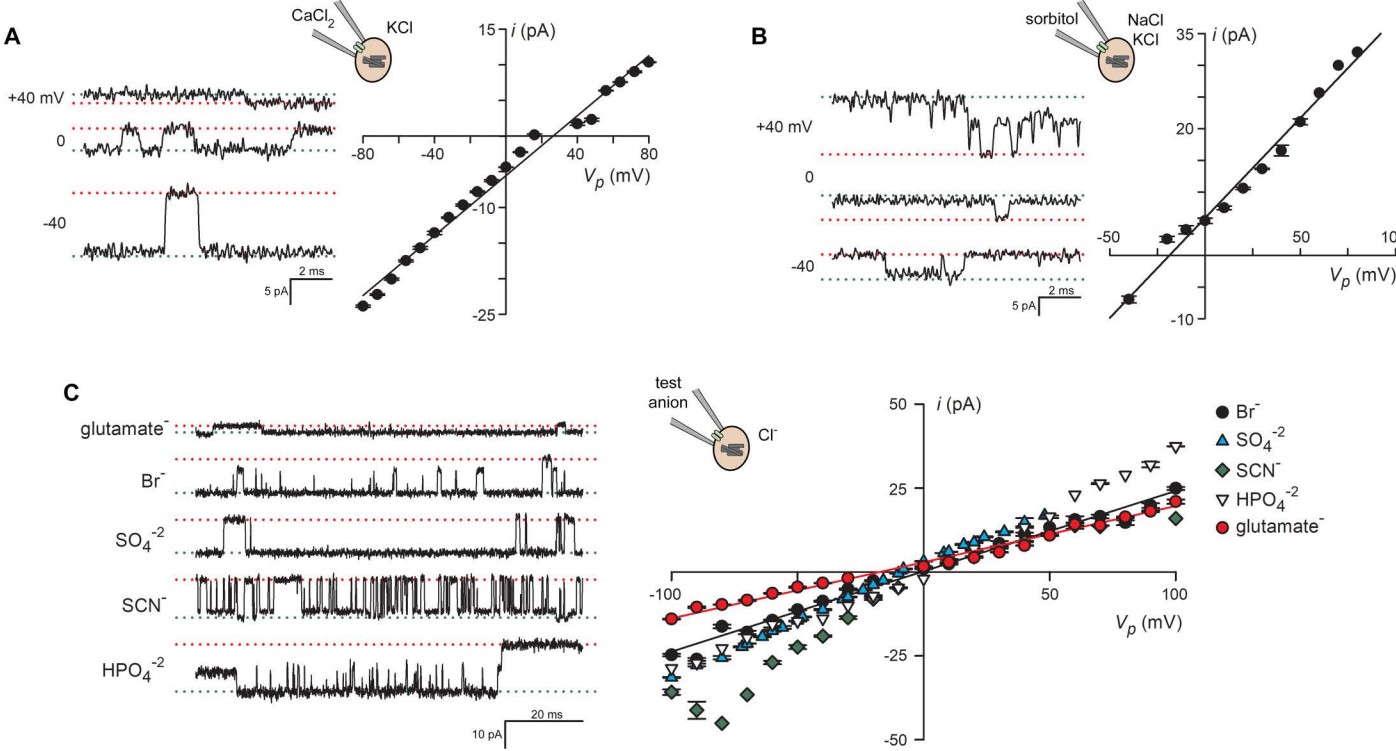

**Fig 2. BVAC is permeant to diverse inorganic and organic anions. (A)** Left, single channel recordings with 100 mM CaCl$_2$ and 50 mM KCl as the primary charge carriers in bath and pipette solutions (Buffers B and C, respectively; S2 Table). $V_p$ as indicated. Notice the negative, downgoing events that reflect channel openings at zero $V_p$ and the smaller amplitudes at positive than equal negative potentials. The cartoon illustrates the recording arrangement. Right, *i-V* relationship showing a positive reversal potential (*x*-intercept), indicating either net anion influx or cation efflux at zero $V_p$. **(B)** Left, single channel recordings with 280 mM sorbitol in the pipette (Buffer D plus WOS) and 70 mM NaCl, 70 mM KCl in the bath (Buffer A plus WOS). Channel events are larger at positive $V_p$ and upgoing at zero voltage. Right, current-voltage relationship showing negative reversal potential establishes preferential anion flux. **(C)** Bi-ionic experiments with indicated anions at 140 mM in the pipette and 140 mM Cl$^-$ in the bath (top to bottom traces pipette/Bath solutions: Buffer E plus WOS/F plus WOS; Buffer G plus WOS/H plus WOS; Buffer I/F; Buffer J plus WOS/F plus WOS; Buffer K/F). $V_p = -60$ mV for all traces. Downgoing channel openings in each solution indicate that each anion is permeant; glutamate$^-$ produces the smallest inward currents, reflecting a lower but nonzero BVAC permeability. Right, *i-V* relationships for these channels; modest shifts in reversal potential from 0 mV indicate relatively high permeability for each anion. The underlying data can be found at https://doi.org/10.5281/zenodo.15305314.

inward currents at negative membrane potentials indicated a moderate flux somewhat lower than seen in experiments with Cl$^-$ as the primary anion; a negative $E_{rev}$ of −17.5 mV corresponds to a $P_{Cl}/P_{glutamate}$ of 2.1 for this channel. As we did not detect intrapore competition between Cl$^-$ and uncharged sorbitol or sucrose, we were unable to assess flux of uncharged or positively charged amino acids. Nevertheless, these selectivity studies reveal broad permeability to organic and inorganic anions of varying net charge and strongly suggest a role in amino acid salvage necessitated by hemoglobin digestion within the DV.

## Effects of pH and Ca$^{++}$ distinguish BVAC from mammalian lysosomal channels

The DV is often compared to mammalian lysosomes as both use acid hydrolases to metabolize macromolecules. As several lysosomal channels are regulated by luminal pH, cytosolic Ca$^{++}$, and other soluble modulators [22], we explored whether BVAC may also be regulated. We were unable to use whole-organelle patch-clamp, which has enabled quantitative studies of lysosomal channel regulation through control of solution compositions at both membrane faces. The DV maintains a large outward H$^+$ gradient through the action of two H$^+$ pumps in both intact infected cells and after separation

from parasite cytosol [23,24]; such gradients are only possible with negligible BVAC permeability to $H^+$ because this channel's large conductance would overwhelm the much lower rates of active extrusion pumps. We therefore examined BVAC activity after luminal alkalinization using treatment with concanamycin A, an inhibitor of the DV's $H^+$ ATPase pump [25], and $NH_4Cl$, a weak base that rapidly increases DV pH [26]. Recordings with these treatments revealed negligible effects on the DV channel's gating, conductance, and open probability (S6 Fig), suggesting that BVAC is not regulated by DV luminal pH. We were, however, unable to quantify pH changes within single DVs. Future studies implementing the whole-organelle or excised patch configurations will permit more rigorous examination of luminal pH regulation as these methods allow precise control of pH at both channel faces.

We then performed patch-clamp using altered pipette solution pH to explore pH effects at the channel's cytoplasmic face (S7 Fig). Here, we found marked changes in gating and conductance, with reduced channel gating at pH 6.0. At pH 8.5, conductance was increased with a preponderance of channels that form functional dimers, as occasionally observed at physiological pH (S3E Fig)

These pH studies establish $H^+$-titratable domains on the channel, as expected for voltage-dependent proteaceous pores. They also distinguish the parasite DV channel from lysosomal channels, many of which are regulated by luminal pH [22]. As changes in cytosolic pH have a greater effect on BVAC behavior than luminal pH changes, these findings suggest greater exposure of $H^+$-titratable residues that the cytosolic surface, possibly as a result of evolution to limit acid denaturation and protease digestion at the luminal channel face.

We also performed patch-clamp with nominally $Ca^{++}$-free solutions to examine regulation by cytosolic $Ca^{++}$, as seen with lysosomal BK channels [27]. These studies revealed unchanged gating, conductance, and open probability (S8 Fig). Thus, while further studies to explore regulation of DV membrane transport are warranted, the lack of physiological regulation by pH and $Ca^{++}$ distinguishes BVAC from lysosomal ion channels.

## A candidate gene strategy excludes contributions from PfCRT and PfMDR1

Along with the limited understanding of DV transport properties, the physiological roles of most known DV membrane proteins remain unclear. PfCRT, a conserved protein that determines resistance to chloroquine, piperaquine, and other antimalarials [28], is thought to export antimalarials from their DV sites of action [29,30]. Its physiological substrates are still debated with proposals including transport of short peptides, amino acids, glutathione, $Fe^{++}$, $Ca^{++}$, and/or $Cl^-$ [11–14,31]. A second DV resident membrane protein, PfMDR1, is homologous to mammalian P-glycoprotein, contributes to antimalarial resistance through mutations and gene copy number variation [32–35], and also has unknown physiological substrates. Importantly, prior studies could not determine whether these putative transport proteins exhibit channel- or carrier-type kinetics [36]. We therefore devised a candidate gene strategy to evaluate possible contributions of PfCRT and PfMDR1 to BVAC activity.

Our strategy is based on detecting altered or reduced BVAC activity upon conditional knockdown of individual candidate genes. We designed a tandem knockdown strategy for PfCRT that combines conditional transcription-level knockdown using the *glmS* riboswitch [37] and posttranslational degradation of the DHFR degradation domain (DDD) C-terminal degron [38] (Figs 3A and S9A). DNA transfection to add a C-terminal HA epitope tag with DDD and a silent *glmS* yielded the *CRT-KD* clone (S9B Fig). Indirect immunofluorescence verified faithful trafficking of modified PfCRT to the DV as well as the expected knockdown upon *glmS* self-cleavage through addition of glucosamine (GlcN) or DDD degradation by removal of trimethoprim (TMP); maximal PfCRT knockdown was seen when both methods were combined (S9C Fig). Immunoblotting confirmed these findings and indicated 95% knockdown in enriched DVs (Fig 3B, 3C). Interestingly, blots using total cell lysates showed a somewhat lower 80% knockdown under our conditions, suggesting retention of mistrafficked, presumably denatured protein after TMP removal. Growth inhibition studies revealed that either TMP removal or a 34 h GlcN treatment of synchronized trophozoite-stage infected cells killed *CRT-KD*, indicating that conditional PfCRT knockdown is lethal (Fig 3D). Growth of the parental Dd2 line was not affected by this GlcN pulse, excluding nonspecific toxicity. Thus, PfCRT knockdown adequately compromised its physiological roles to abort parasite growth.

 

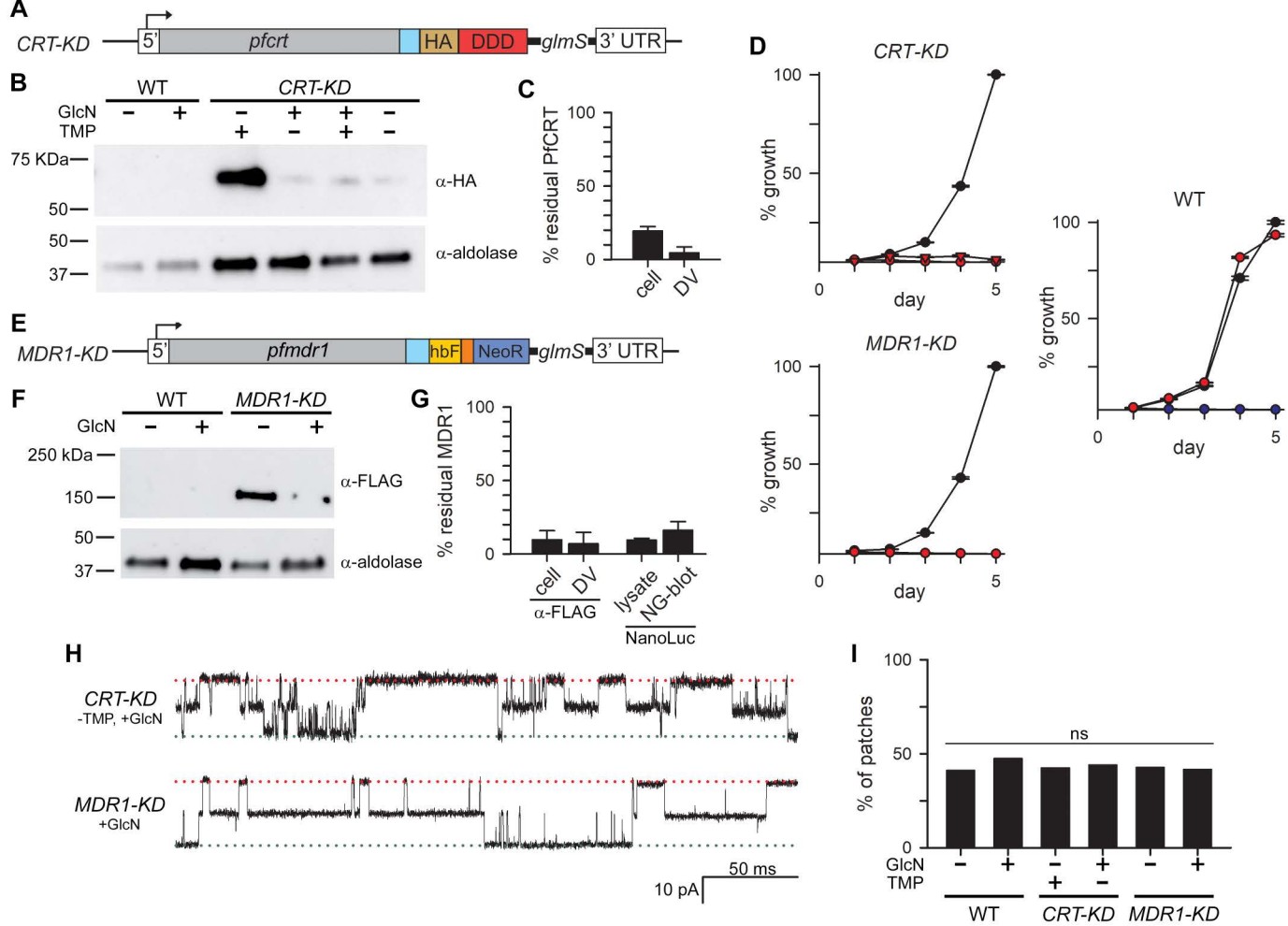

**Fig 3. Conditional knockdowns establish essentiality of PfCRT and PfMDR1 and exclude links to BVAC. (A)** Ribbon schematic showing the modified *pfcrt* gene carrying a linker (blue), hemagglutinin tag (HA), DHFR degradation domain (DDD), and a 3′ untranslated *glmS* riboswitch. Removal of trimethoprim (TMP) destabilizes DDD and degrades PfCRT; glucosamine (GlcN) addition degrades the gene's mRNA. **(B)** Immunoblot showing conditional knockdown of PfCRT, as quantified using isolated DVs and anti-HA antibody. Addition of GlcN or removal of TMP reduces PfCRT; together, these manipulations maximize knockdown. The wild-type negative control is included (WT). Bottom, aldolase loading control. **(C)** Mean ± S.E.M. % residual PfCRT upon tandem knockdown after loading control normalization, from total cell lysates (cell) or isolated DVs (DV). **(D)** Growth of indicated parasites over 5 days. Parental wildtype line (WT): control (black); GlcN pulse addition (red); chloroquine growth inhibition control (blue). *CRT-KD*: medium + TMP (black); TMP removal (red triangles); GlcN pulse (red circles). *MDR1-KD*: control medium (black); GlcN pulse (red circles). PfCRT or PfMDR1 knockdown abolishes parasite expansion ($P < 0.05$ for TMP removal or GlcN prepulse for *CRT-KD*, $n = 4$ independent trials; $P < 0.05$ for GlcN prepulse for *MDR1-KD*, $n = 3$). **(E)** Schematic of modified *pfmdr1* with linker (blue), tandem HiBiT and FLAG tags (hbF), T2A ribosomal skip peptide (orange), a neomycin resistance gene (NeoR), and a 3′ untranslated *glmS*. **(F)** Anti-FLAG immunoblot showing PfMDR1 knockdown upon GlcN pulse treatment. The wild-type control and aldolase loading controls are included. **(G)** Mean ± S.E.M. % residual PfMDR1 after knockdown. Total membranes (cell) or enriched DVs (DV) were estimated from anti-FLAG immunoblots. NanoLuc luminescence was quantified using total cell lysates (lysate); PfMDR1 abundance was also measured using Nano-Glo blotting of total membranes after SDS-PAGE (NG-blot). **(H)** Single channel recordings after maximal knockdown of PfCRT or PfMDR1, as indicated to the left of each trace. Symmetric pipette and bath solutions: Buffer A with and without WOS (top and bottom traces); $V_p = -60$ mV. Channel activity was unchanged; both channels shown here exhibit functional dimerization as described in S3E Fig. **(I)** % of patches containing 1 or more channels for indicated parasites and treatments. PfCRT or PfMDR1 knockdown does not change BVAC abundance. ns, no statistically significant difference ($P \geq 0.69$ for all comparisons, central Fisher's exact test with melding confidence intervals) [67]. The underlying data can be found at https://doi.org/10.5281/zenodo.15305314.

Attempts to produce a similar tandem knockdown of PfMDR1 were unsuccessful, possibly because the C-terminal HA-DDD tag is not tolerated on this transporter. We therefore produced a line, *MDR1-KD*, with the *glmS* riboswitch for PfMDR1 knockdown; selection-linked integration with the *neomycin* resistance selection marker facilitated recovery (Figs 3E and S10A–S10C). While studies have implicated *pfmdr1* gene duplication in Dd2 to increase transporter abundance and achieve drug resistance [39], southern blotting with our Dd2 parental line and derivative *MDR1-KD* indicated a single copy that was faithfully modified (S10D Fig); *pfmdr1* copy number is known to vary in parasite culture with fitness cost driving copy number reduction in the absence of sustained selective pressure [40]. Indirect immunofluorescence and immunoblotting studies confirmed DV localization of modified PfMDR1 and the desired GlcN-mediated knockdown (Figs 3F, 3G and S10E, S10F), with 7%–17% residual protein depending on the probing method. As with PfCRT, our growth inhibition studies established that knockdown adequately compromises PfMDR1 essential roles and abolishes parasite expansion (Fig 3D).

Because the precise mechanisms by which *pfmdr1* gene duplication confers antimalarial resistance are unknown, multidrug-resistant clones such as Dd2 may carry changes in associated but uncharacterized DV transporters; supporting this, human MDR1 has been linked to the activity of a distinct Cl⁻ channel [41]. We therefore performed similar PfMDR1 knockdown studies in the drug-sensitive 3D7 background (*MDR1-KD*$_{3D7}$ line, S11 Fig).

We then harvested DVs from our PfCRT and PfMDR1 transfectants after maximal knockdown, performed single-channel studies and found unaltered BVAC behavior (Figs 3H and S11G). Because a reduced number of unaltered channel molecules may persist despite 90%–95% knockdown of an associated protein, we measured the fraction of membrane patches with channels and found that 48 out of 108 *CRT-KD* patches (44%), 29 of 69 *MDR1-KD* patches (42%) and 16 of 34 *MDR1-KD*$_{3D7}$ patches (47%) had one or more functional channels. These abundance rates were statistically indistinguishable from both the corresponding parental lines and the engineered lines without conditional knockdown (Figs 3I and S11H). Thus, neither PfCRT nor PfMDR1 contributes to BVAC formation or regulation.

## Unexploited pharmacology

We assessed possible effects of antimalarial drugs, inhibitors, and transport modulators on BVAC activity (Fig 4). When added to both pipette and bath solutions, the approved antimalarials chloroquine, mefloquine, or artemisinin did not compromise BVAC-mediated open probability at $V_p$ of −20 mV, a membrane potential that allows sensitive detection of inhibition because the channel is primarily open. Cyclosporin A and XR-9576, two P-glycoprotein inhibitors that directly or indirectly block PfMDR1-mediated transport of reporter solutes into DVs [42,43], and verapamil, a Ca⁺⁺ channel blocker that modulates chloroquine susceptibility by poorly understood effects on the DV, also did not reduce BVAC open probability. Several nonspecific inhibitors of anion channels in other organisms had modest or no effect on Cl⁻ flux (Fig 4). Each of these compounds was used at levels above its parasite culture $IC_{50}$ value to evaluate partial contributions to DV physiology. None of these agents produced unambiguous changes in channel gating (S12 Fig) at larger applied voltages, but weak interactions with the channel could not be excluded by our studies. Thus, BVAC is a novel, unexploited antimalarial drug target.

## Discussion

We identified an unusual large conductance, anion-selective channel as the primary conductive transport activity on rapidly harvested, metabolically active DVs from the virulent human malaria parasite, *P. falciparum*. This channel, termed BVAC, is abundant on the DV membrane, constitutively open at the resting membrane potential, and permeant to a broad range of anions. Although we established preferential flux of anions, our selectivity studies could not exclude lower-level cation transport because single-channel patch-clamp cannot exclude low-level flux of ions. BVAC must exclude H⁺ because measured single channel currents are 10⁶-fold greater than H⁺ pump rates [44]; even a low H⁺ permeability (*e.g.*, $P_{H+}/P_{Cl-}$ of 0.01) would abolish the pH gradient across the DV membrane, compromising the DV acidic environment,

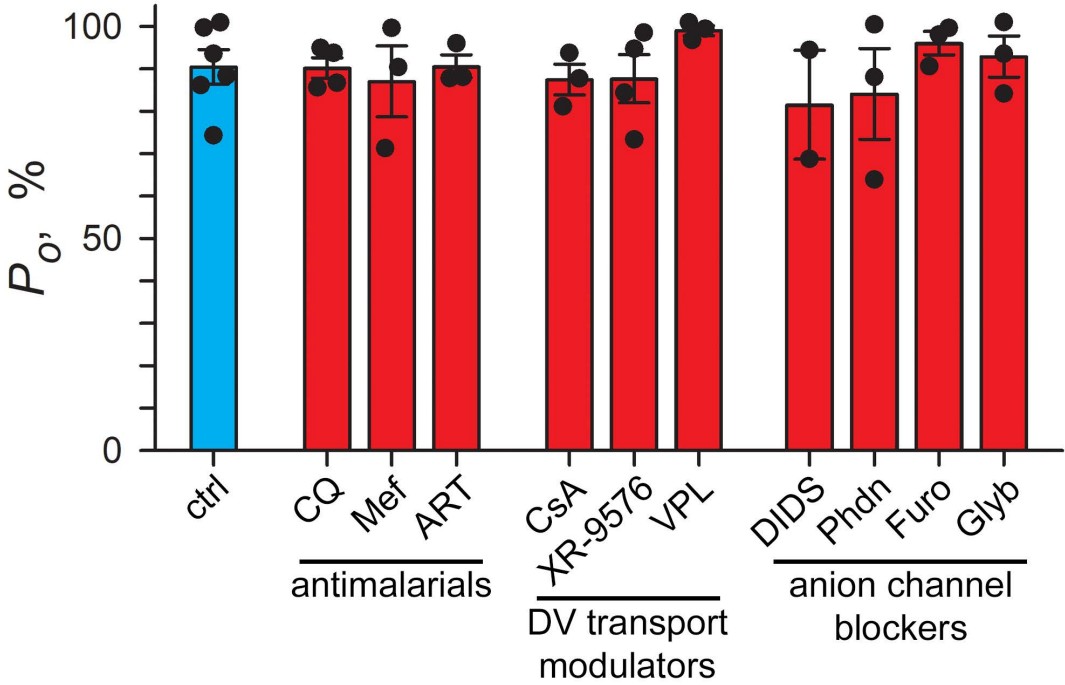

**Fig 4. An unexploited drug target.** Single channel open probabilities ($P_o$) at $V_p$ of −20 mV without (ctrl) and with inhibitors present in both bath and pipette: CQ, 1 μM chloroquine; Mef, 1 μM mefloquine; ART, 1 μM artemisinin; CsA, 10 μM cyclosporin A; 3 μM XR-9576; VPL, 20 μM verapamil; 200 μM DIDS; Phdn, 100 μM phloridzin; Furo, 100 μM furosemide; Glyb, 200 μM glybenclamide. Pipette and bath solutions of Buffer A with WOS. Filled circles reflect single molecule recordings used for analysis. The underlying data can be found at https://doi.org/10.5281/zenodo.15305314.

hemoglobin digestion and heme detoxification. Water and cation channels are also known to exclude $H^+$ to sustain pH gradients at diverse membranes [45,46]. A $Ca^{++}$ gradient at the DV membrane established by putative ATPase pumps also implicates negligible passive $Ca^{++}$ flux through BVAC [47].

We propose that BVAC serves two essential roles (Fig 5). Its high $Cl^-$ permeability ensures that active $H^+$ import via V-type ATPase and pyrophosphatase pumps [23,25] is matched stoichiometrically by $Cl^-$ uptake, maintaining electro-neutrality within the DV. Without BVAC, DV acidification would be halted due to excessive charge buildup and an overly negative DV membrane potential (Eq. 1). BVAC's permeability to glutamate suggests a second role in export of negatively charged amino acids liberated by hemoglobin digestion. Although our recording conditions did not permit examination, BVAC may also be permeant to neutral and/or positively charged amino acids and oligopeptides generated in the DV [5]. As these hemoglobin digestion products are needed to fuel parasite protein translation, BVAC inhibition may lead to parasite death by two distinct mechanisms.

Our studies highlight the advantages of direct measurements on single isolated DVs: these mechanistic insights have eluded prior studies that were limited to solute accumulation measurements using semi-enriched DV preparations. Macroscopic measurements using such preparations are plagued with contributions from unwanted cellular compartments and organelles. Because we measured transporter activity in native DV membranes, our findings also avoided concerns associated with heterologous expression systems that may not faithfully reproduce transporter behavior [48]. Finally, we also found chemical pretreatment to enlarge DV size unnecessary for patch-clamp; this is in contrast to mammalian lysosome studies that routinely use pretreatment with vacuolin, a chemical that artificially increases lysosome size by unknown mechanisms [49,50].

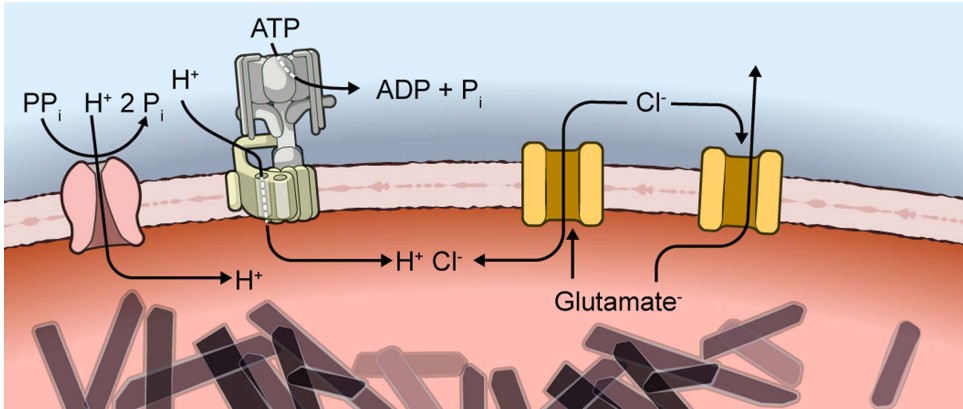

**Fig 5. Model showing two essential roles for BVAC.** Anion uptake via BVAC (yellow) maintains electroneutrality during DV acidification via active $H^+$ uptake through pyrophosphatase and V-type ATPase pumps (pink and green transporters, respectively). $Glutamate^-$ liberated by hemoglobin digestion is exported via BVAC, fueling parasite protein synthesis; the channel may also export neutral and positively charged amino acids.

Channels other than BVAC were not detected in the various solutions used in our recordings. This finding distinguishes the DV from mammalian lysosomes, where various ion channels for $K^+$, $Na^+$, $Ca^{++}$, and $Cl^-$ have been identified [22]. This paucity of channels parallels the absence of clear bacterial and vertebrate ion channel orthologs in the parasite genome; it also raises questions about the molecular basis of BVAC. Candidate gene knockdown (Figs 3 and S9–S11) represents a powerful, agnostic approach for addressing this unknown. A single published proteome of the DV reveals a number of hypothetical membrane proteins with multiple transmembrane domains [51], which should be evaluated in future studies.

Multiple antimalarial drugs act within the DV, an organelle often considered the metabolic hub of the bloodstream pathogen. Chloroquine, piperaquine, and their 4-aminoquinoline derivatives interfere with heme polymerization within the DV. Acquired resistance to these once highly effective antimalarials has been linked to PfCRT with lesser contributions by PfMDR1 and other putative transporters on the DV membrane [10,28,34]. Mefloquine also acts within the DV, with acquired resistance linked to PfMDR1 [8,9]. Efflux of unmetabolized antimalarial drugs via PfCRT and/or PfMDR1 is the currently preferred resistance mechanism, but this remains to be conclusively established. Although intensively studied, whether these transporters exhibit channel-versus carrier-type kinetics is unclear and their physiological substrates remain debated with amino acids, short peptides, other organic solutes, and inorganic ions still under consideration. Our studies largely exclude channel-type mechanisms for PfCRT and PfMDR1 because their knockdowns did not produce changes in membrane conductance in organelle-attached patch-clamp. Whole-DV patch-clamp, as already achieved for mammalian lysosomes [49], may eventually uncover the in situ roles of these transporters if combined with our conditional knockdown strategy.

Direct electrical measurements on the DV membrane enable foundational insights into transport of inorganic and organic solutes into and out of the parasite's metabolic hub and should permit the development of novel antimalarial drugs with defined mechanisms of action.

## Methods

### Parasite culture

*P. falciparum* laboratory clones and their engineered derivatives were cultivated at 37 °C under 5% $O_2$, 5% $CO_2$, 90% $N_2$ in $O^+$ human erythrocytes (University of Virginia Blood Bank) at 5% hematocrit in RPMI 1640-based medium supplemented with 25 mM HEPES, 50 µg/mL hypoxanthine (KD Medical), 0.5% w/v Albumax II (Gibco), 1 µg/mL Gentamicin and 28.6 mM $NaHCO_3$ (KD Medical). The Dd2 clone, originally from Indochina, is resistant to antimalarials including chloroquine, mefloquine, and quinine, which target the DV; 3D7 and HB3 are sensitive to these agents [52–54].

## Digestive vacuole harvest and enrichment

We modified an existing protocol for DV harvest and enrichment [55] to prioritize rapid harvest and organelle viability over purity because individual DVs can be visually identified for patch-clamp. Trophozoite-stage *P. falciparum* cultures at ~5% parasitemia were enriched by Percoll-sorbitol density gradient centrifugation, washed in HBS (137 mM NaCl, 2.7 mM KCl, 20 mM HEPES, pH 7.4) and lysed in HBS with 0.5 µg/mL saponin. Freed parasites were washed twice with chilled HBS (3,000 *g*, 20 s), resuspended in 10 volumes chilled 20 mM MES, pH 4.5. DVs were released by trituration through a 1.2 cm, 27-G needle four times and pelleted at 21,000 *g* at 4 °C for 2 min. The pellet was resuspended in Na-K buffer (70 mM KCl, 70 mM NaCl, 2 mM $CaCl_2$, 2.5 mM $MgCl_2$, 10 mM HEPES, 10 mM MES, pH 7.4), mixed with 80% Percoll-sucrose (80% v/v Percoll, 0.25 M Sucrose, 1.5 mM $MgSO_4$), triturated twice through a 27-G 1.2 cm needle and centrifuged (21,000 *g* at 4 °C for 2 min). DVs, enriched in a dark floating band near the bottom of the gradient, were resuspended in Na-K buffer supplemented with 1 mM $Na_2ATP$, 0.3 mM $Na_2GTP$, 8.8 $Na_2$-phosphocreatine (Sigma Aldrich), and stored on ice until use. Microscopic examination and membrane fluorescence in the Dd2[attB]/CRT-GFP line [56] confirmed DV integrity; active hemozoin tumbling within the DV was indicative of fresh harvest.

When used, detergent-free harvest of DVs was performed identically without pre-lysis in HBS with saponin.

## Electrophysiology

Freshly harvested DVs were used for organelle-attached patch-clamp under voltage-clamp conditions. Pipettes with narrow tips and acceptably low access resistances (7–13 MΩ with isotonic salt solutions) were freshly pulled from KG-33 capillary glass (1.3 mm OD, 0.7 mm ID, King Precision) using a two-line program (P-1000, Sutter Instruments). The pipette diameter proximal to the tip was expanded by heating in a microforge with internal positive pressure (0.1 psi, heat setting at 80%, DMF1000 microforge controller, World Precision Instruments), reducing access resistance and facilitating filling. With practice and optimized puller settings, pipettes with similar tips and low access resistances could be prepared without shank heat expansion. Pipettes were coated with Sylgard 184 silicone elastomer (Dow Corning), fire-polished, filled with salt solution, and used immediately for organelle patch-clamp at room temperature. After gigaseal formation, recordings were obtained using an Axopatch 200B amplifier (Molecular Devices) with 0 mV holding potential. Data were low-pass filtered at 5 kHz (8-pole Bessel, Frequency Devices), digitized at 100 kHz (Digidata 1550A), and recorded with Clampex 10.7 software (Molecular Devices).

We followed the voltage sign convention proposed for endomembranes [57], so that the DV membrane potential ($V_m$) is given by:

$$V_m = V_{cytosol} - V_{DV\ lumen} \tag{1}$$

Consistent with this convention, positive currents in single channel traces and IVs reflect anion flow from the DV lumen into the cytosol or pipette solution in organelle-attached patch-clamp; negative currents reflect anion flow into the DV. Recordings are presented at the applied pipette potential ($V_p$), which scales linearly with ambient $V_m$. Changes in the reversal potential, $E_{rev}$, reflect relative permeabilities of ions in the pipette and bath solutions.

Single-channel analysis was carried out using Clampfit (v10.7, Molecular Devices) and locally developed code (DIAdem 2015, National Instruments; SigmaPlot 14.5, Systat Software). Channel open and closed levels were determined manually by visual examination of multiple traces to ensure faithful detection of primary conductance levels and avoid misassignment to long-lived subconductance levels. Channel open probabilities were estimated by integration of single channel currents after subtraction of closed channel leak currents, which were ohmic over the applied voltage range. Open probability-voltage plots were fitted to the Boltzmann equation:

$$\frac{i}{i_{max}} = \left[ \frac{1}{\left(1 + exp\left(\frac{E_{0.5} - V_p}{k}\right)\right)} \right] + a \tag{2}$$

where $i$ and $i_{max}$ are the time-integrated single channel currents at $V_p$ and saturating currents at near-zero imposed potentials, respectively. $E_{0.5}$ is the potential producing a half-maximal current; $k$ and $a$ are the slope factor and offset values. Data at positive and negative $V_p$ were fitted separately.

Differing bath and pipette solutions were used to examine the channel's selectivity for ions. The channel's reversal potential, $E_{rev}$, was then measured by linear interpolation of current amplitudes at nearby applied $V_p$. Experiments using sorbitol as the primary osmoticant in the pipette (Buffer D, S2 Table) were used to estimate the channel's preference for anions using:

$$\frac{P_{Cl}}{P_{cation}} = \frac{[cation]_{bath} - \{10^{\left(-E_{rev} * \frac{F}{RT}\right)} * [cation]_{pipette}\}}{\{10^{\left(-E_{rev} * \frac{F}{RT}\right)} * [Cl]_{bath}\} - [Cl]_{pipette}} .$$

(3)

This equation represents a rearrangement of the Goldman equation [58] and excludes contributions of divalent ions. Here, [cation] is the sum of $Na^+$ and $K^+$ concentrations while $F, R,$ and $T$ have their conventional definitions.

### Live cell imaging

Hemozoin movement within infected cells or enriched DVs was used to evaluate DV metabolic activity and was imaged at room temperature. Imaging was performed using a Nikon Eclipse Ti2-U inverted microscope with a 100× oil immersion objective. A OnePlus 7 Pro with a 48 MP camera was mounted on the microscope eyepiece using a smartphone adaptor (Gosky) to record videos before and after DV capture on a patch-clamp pipette. Room temperature incubation of enriched DVs was used to visualize the loss of metabolic activity. All recordings were uniformly magnified and trimmed to required length using an online video editor (Clipchamp).

Live cell imaging of GFP fluorescence in the Dd2attB/CRT-GFP parasite was performed on a Nikon AX confocal microscope using a 60× oil emersion objective with excitation and emission at 488/20 nm and 530/25 nm). DV hemozoin movement was recorded with an imaging speed of 30 frames per second. Data was denoised using denoise.ai build with MXNET framework (Nikon).

DVs harvested from the plasmepsin II-E(1)GFP parasite were imaged using a Zeiss LSM 880 confocal microscope using a Zeiss plan apochromat 63×/1.4 NA oil immersion objective. Samples were excited at 405 nm with a dwell time of 1.03 µs, average of 4, and pinhole of 1.75 AU. Fluorescence was detected from 506 to 600 nm using a PMT with gain 650 and offset 750. Although the E(1)GFP variant is engineered to permit ratiometric pH measurements in nascent lysosomes [59], our studies revealed that this reporter was unresponsive to changes in DV pH, possibly due to denaturation by the more acidic pH of DVs or to partial digestion in this organelle.

### Transmission electron microscopy

Enriched DVs were fixed overnight in 0.1 M sodium cacodylate with 2.5% glutaraldehyde, washed, and transported on ice. Samples were washed three times with 0.1 M sodium cacodylate buffer, then postfixed with 0.5% $OsO_4$–0.8% $K_4Fe(CN)_6$ in 0.1 M sodium cacodylate for two times (2 min on-2 min off-2 min on cycle) at 80 W/24 °C under vacuum at 20 in. Hg in a Pelco Biowave laboratory microwave system (Ted Pella, Redding, CA). After postfixation, the samples were washed twice in 0.1 M sodium cacodylate at 250 W/24 °C without vacuum. The cells were then stained twice with 1% tannic acid in distilled water ($dH_2O$) (2 min on-2 min off-2 min on cycle) at 80 W/24 °C under 20 in. Hg vacuum. The samples were then rinsed twice with $dH_2O$ for 45 s at 250 W/24 °C without vacuum. Following the rinses, the samples were stained twice with 2% samarium acetate in $dH_2O$ (2 min on-2 min off-2 min on cycle) at 80 W/24 °C under 20 in. Hg vacuum. The samples were dehydrated in a graded ethanol series followed by three exchanges in 100% ethanol. Following dehydration, the samples were infiltrated and embedded with Spurr's resin and cured overnight at 68 °C, sectioned at 70 nm using a UC6 ultramicrotome (Leica Microsystems), and viewed on a Tecnai BioTwin Spirit (Thermo Fisher Scientific) at 120 kV. Images were acquired with a Hamamatsu ORCA-HR digital camera system (Advanced Microscopy Techniques).

## Plasmid construction and DNA transfection

Parasite DNA transfections were performed using CRISPR-Cas9 gene editing to generate epitope-tagged PfCRT and PfMDR1 conditional knockdown lines as well as plasmepsin II-E(1)GFP, a soluble DV reporter line. Gene editing utilized the standard pUF1-Cas9 and modified pL7 plasmids [60]. The sgRNA were selected using on- and off-target scores as tabulated for *P. falciparum* [61]. Modified pL7 plasmids were constructed through restriction digestion and ligation to introduce synthetic double-stranded DNA (gBlocks, Integrated DNA Technologies) and PCR amplicons from genomic DNA. To promote retention of successful integrants, shield mutations were introduced at the sgRNA binding site on the donor sequence used for homology-directed repair. In-Fusion cloning (Takara Bio) was used to clone complementary primers that encode the sgRNA, as listed in S1 Table.

The PfCRT knockdown was produced using the multidrug-resistant Dd2 laboratory clone; PfMDR1 knockdown used both Dd2 and the drug-sensitive 3D7 line because of possible effects of *pfmdr1* copy number variation. For both genes, enriched trophozoite-stage infected cells were cultivated in uninfected plasmid-loaded human erythrocytes for 48 h prior to selection with 2.5 nM WR99210 and 1.5 μM DSM1. The PfCRT knockdown was additionally supplemented with 20 μM TMP (Sigma Aldrich). Integration was confirmed by PCR after parasite outgrowth and luminescence measurements for PfMDR1 integrants carrying the HiBiT epitope tag (Promega). PCR primers used are tallied in S1 Table. All experiments were performed with limiting dilution clones after confirmation with DNA sequencing. Validated clones were maintained without drug selection except for *CRT-KD*, which was maintained with 20 μM TMP for normal PfCRT expression.

PfCRT was epitope-tagged with a C-terminal 3xHA-tag followed by a DDD [38], stop codon, and the *glmS* ribozyme [37], permitting both transcription- and post-translational conditional knockdown. Because PfMDR1 was refractory to addition of this large C-terminal tag in our experiments, we produced conditional knockdown using a smaller C-terminal tandem Hi-Bit and 1x FLAG epitope tag followed by the T2A skip peptide [62], the neomycin resistance gene (NeoR) [63], a stop codon and the *glmS* ribozyme. G418 sulfate (400 μg/ml, Gibco) was added for 1–2 weeks after parasite outgrowth to select for NeoR expression and facilitate recovery of the PfMDR1 integrant.

## Conditional knockdown of candidate DV membrane transporters

Conditional knockdown of PfCRT and PfMDR1 was used to evaluate contributions to BVAC activity. Knockdown was individualized for each candidate transporter to maximize knockdown and recovery of DVs for biochemical studies. The *CRT-KD* PfCRT clone was routinely maintained in standard culture medium with 20 μM TMP to preserve PfCRT function. Knockdown was initiated using synchronous schizont-stage cultures by removal of TMP and/or addition of 8 mM glucosamine HCl (GlcN, MP Biomedicals). After cultivation for 34 h, microscopically confirmed trophozoite-stage infected cells were harvested and used for knockdown quantification and biochemical studies. PfMDR1 conditional knockdown was achieved by cultivation of synchronous schizont-infected cultures with 4 mM GlcN for 36 h. Off-target GlcN toxicity was evaluated with identical treatment of the corresponding parental clones.

## Protein blots

Immunoblotting and Nano-Glo HiBit blotting were used to quantify residual protein abundance after conditional transporter knockdown. These experiments used either enriched DVs or total cellular membranes from Percoll-sorbitol-enriched trophozoite-infected cells. The total membrane fraction was prepared by hypotonic lysis in 7.5 mM $Na_2HPO_4$, 1 mM EDTA, pH 7.5, and ultracentrifugation (150,000 g, 4 °C, 1 h). Matched samples were solubilized in Laemmli sample buffer containing 6% SDS, separated by electrophoresis in 4–15% Mini-PROTEAN TGX gels (Bio-Rad) and transferred to nitrocellulose membranes.

For immunoblots, the nitrocellulose membrane was blocked (5% skim milk powder in TBST: 150 mM NaCl, 20 mM Tris-HCl, 0.1% Tween20, pH 7.4) at room temperature for 1 h before addition of primary antibody at a 1:2,000–1:3,000 dilution in blocking buffer for overnight 4 °C incubation with rocking. After three TBST washes, horse radish peroxidase

(HRP)-conjugated secondary antibody was added at a 1:3,000 dilution for 1 h with rocking. After three TBST washes, blots were imaged with Clarity Western ECL substrate (Bio-Rad) and an Amersham 680 imager (GE healthcare). All immuno-blots are representative of at least three independent trials. Band intensities were quantified using ImageJ software (NIH) with aldolase as a loading control (HRP anti-Plasmodium aldolase antibody, Abcam).

Nano-Glo blots (HiBiT blotting system, Promega) were also performed to quantify PfMDR1 knockdown. After SDS-PAGE and transfer to nitrocellulose membranes, blots were washed in 150 mM NaCl, 20 mM Tris-HCl (pH 7.4) with 0.1% Tween 20. LgBiT was applied at a 1:200 dilution with overnight 4 °C incubation with rocking. Furimazine was then added in blotting buffer at a 1:500 dilution before imaging and band intensity quantification as above.

## Indirect immunofluorescence microscopy

Indirect immunofluorescence assays (IFA) were performed using freshly prepared thin blood smears after fixation with 1:1 acetone: methanol at −20 °C for 5 min or 4% paraformaldehyde with 0.1% Triton X-100 in PBS. After air drying, slides were blocked with 5% non-fat milk in PBS for 1 h at RT and incubated with primary antibody at a 1:500 dilution in blocking buffer (mouse anti-HA or mouse anti-FLAG M2 antibody, Sigma) for 2 h at RT under a coverslip. After three chilled PBS washes, Alexa Fluor 488-conjugated secondary antibody (1:1000; Invitrogen) with 1 µg/mL 4′,6-diamidino-2-phenylindole (DAPI; Molecular Probes) was added in blocking buffer and incubated for 1 h at RT. After chilled PBS washes, slides were dried and mounted using ProLong Diamond Antifade Mountant (Molecular Probes). Images were collected on a Leica SP8 microscope using a 63× oil immersion objective. DAPI and Alexa Fluor 488 fluorescence were captured using excitation/emission wavelengths of 377/447 nm and 493/519 nm, respectively. Images, representative of >500 cells from at least 50 fields each, were processed using Leica LAS X and Huygens Essential software.

## Growth inhibition assays

The effects of candidate transporter knockdown on parasite viability were quantified using SYBR Green I fluorescence assays to measure parasite nucleic acid production [64]. Synchronous trophozoite-stage cultures were seeded at 0.5% parasitemia, 5% hematocrit in standard media with or without TMP and GlcN to produce PfCRT or PfMDR1 knockdown; GlcN was continued for optimized durations described above. Matched cultures of the engineered line without knockdown as well as its corresponding parent were seeded to compare growth kinetics and assess GlcN toxicity. Chloroquine (20 µM) was included as a growth inhibition control. Medium was changed daily; matched aliquots were harvested daily and frozen. After completion, frozen cell pellets were lysed in 20 mM Tris, 10 mM EDTA, 0.016% saponin, 1.6% Triton X-100, pH 7.5 with a 5,000× dilution of SYBR Green I nucleic acid gel stain (Thermo Fisher Scientific) and transferred to a 96 well microplate. After a 2 h RT incubation without light exposure, fluorescence was used to quantify parasite nucleic acid (excitation/emission, 485/528 nm; Synergy Neo2, BioTek). The results are representative of at least three independent experiments for each line and are shown as mean ± S.E.M. of triplicate wells after normalization of the matched control to 100% growth on day 5.

## Southern blot

PfMDR1 copy number and replacement with the knockdown cassette were evaluated using Southern blotting. Genomic DNA was extracted using the QIAamp DNA blood mini kit (QIAGEN), digested with indicated enzymes, resolved on a 0.8% agarose gel, acid depurinated, and transferred overnight onto a positively charged nylon membrane (Roche). A digoxigenin (DIG)-labeled 271 bp DNA probe complementary to a *pfmdr1* region within the transfection homology cassette recognizing both wildtype and integrant sequences was produced by PCR amplification of genomic DNA (primers listed in S1 Table). After prehybridization of the cross-linked membrane with DIG-Easy Hyb buffer (Roche), the labeled probe was added and hybridized overnight at 37 °C. The blot was washed sequentially with low-stringency buffer (2 × SSC + 0.1% SDS) at RT and high-stringency buffer (0.1 × SSC + 0.1% SDS) at 50 °C; 1 × SSC is 0.15 M NaCl + 0.015 M sodium

citrate. After blocking, binding was detected using anti-digoxigenin alkaline phosphatase (AP) Fab fragments at a dilution of 1:10,000 and CDP-Star substrate (Roche).

## Data analysis

Statistical analysis of the data was performed using SigmaPlot 14.5 and GraphPad Prism 9.4. The data are reported as individual values and mean$\pm$S.E.M. Comparisons of data were made using the Student $t$ test for two groups or one-way ANOVA for comparisons of more than two groups. The central Fisher's exact test with melding confidence intervals was used for comparisons of single channel abundance comparisons between groups of electrophysiological datasets. Details of the statistical tests applied are provided in the relevant figure legends.

## Supporting information

**S1 Fig. Identification of individual DVs using GFP-tagged PfCRT.** Schematic shows PfCRT transmembrane topology with residues indicated by individual dots (adapted from [65]); GFP (green ribbon) is predicted to localize in parasite cytosol. Confocal fluorescence images showing intact infected erythrocytes or isolated DV expressing GFP-tagged PfCRT, a DV membrane protein marker. Excitation/emission: $485\pm20$ nm/$530\pm25$ nm. Scale bars, 5 µm.
(TIF)

**S2 Fig. Preserved DV membrane integrity during harvest.** Confocal fluorescence, brightfield and merge images showing three separate DVs, harvested the plasmepsin II-E(1)GFP reporter line. Retention of this soluble protein indicates preserved DV integrity. Scale bars, 2 µm.
(TIF)

**S3 Fig. Complex channel gating independent of recording conditions. (A)** Single channel recordings at indicated $V_p$ using DV harvest without saponin; pipette and bath solution: Buffer A with WOS. Unchanged channel activity excludes artifacts associated with detergent exposure. Closed and open channel levels, red and green dashes. **(B)** Recordings without glucose, ATP, GTP, or phosphocreatine in the bath or pipette solutions (Buffer A). Channel activity does not require external energy sources. **(C)** Complex channel gating with multiple subconductance levels, apparent as current levels intermediate between the fully closed and fully open levels marked by red and green dashes. Buffer A with WOS in pipette and bath. Horizontal scale bar: 475.4, 17.0, and 46.8 ms (top to bottom traces, respectively). We excluded independent small-conductance channels in these recordings because of clear interconversions between all current levels; such interconversions are not observed with independent channels. **(D)** Single channel recording showing a functional channel dimer, recorded with Buffer A plus WOS in pipette and bath. Two separate channels in the patch are excluded as there are frequent transitions between both pores closed and both pores open, as marked with red down arrows. The current level corresponding to a single open pore is marked with a dashed blue line. Schematic at right shows a possible structural model for this behavior, where closing of the master gate (blue) could produce the observed double transitions (red down arrow); individual pore gates (orange) account for transitions to and from the dashed blue line. The underlying data can be found at https://doi.org/10.5281/zenodo.15305314.
(TIF)

**S4 Fig. Extended single channel recordings confirm a stable seal on the DV membrane.** Consecutive 500 ms traces recorded with Buffer A in bath and pipette. Upgoing transitions reflect channel closings. Notice the bursts of complex gating mixed with periods of long closings and openings; subconductance levels are also apparent in many sweeps. $V_p=-60$ mV. Scale bar at bottom right, 25 pA/50 ms. The underlying data can be found at https://doi.org/10.5281/zenodo.15305314.
(TIF)

**S5 Fig. Conserved properties on strains having divergent geographies and antimalarial susceptibilities. (A, D)** Single channel recordings on DVs from 3D7 and HB3 parasites, respectively, using Buffer A plus WOS. Notice the conserved single-channel amplitudes, gating, and subconductance levels. $V_p$, −80 mV for both traces. Closed and open channel levels, red and green dashes. **(B, E)** Current-voltage relationships for 3D7 and HB3 channel molecules. **(C, F)** Open probabilities at imposed $V_p$ for 3D7 and HB3 channels. Solid lines, best fit to Eq. 2. There is greater molecule-to-molecule variation within a single strain than between these geographically divergent strains with distinct antimalarial susceptibilities. The underlying data can be found at https://doi.org/10.5281/zenodo.15305314.
(TIF)

**S6 Fig. Negligible effects of DV luminal alkalization by concanamycin A and NH₄Cl. (A)** Single BVAC recordings with Buffer A in pipette and Buffer L in bath. $V_p$ as indicated; closed and open channel current levels, red and green dashes; scale bar, 10 pA/50 ms. **(B)** Current–voltage ($i$–V) relationship. The single-channel conductance, 272 pS, is not significantly affected by luminal alkalization. **(C)** Open probability vs. $V_p$ plot. Solid line, best fit to Eq. 2. The underlying data can be found at https://doi.org/10.5281/zenodo.15305314.
(TIF)

**S7 Fig. Marked effects of exofacial pH on BVAC gating and conductance. (A)** Single BVAC recordings with Buffer A in pipette after adjusting pH to 6.0 or 8.5 pipette (left and right traces); Buffer A in bath. Notice reduced gating at pH 6.0 and functional dimerization at pH 8.5. $V_p$ as indicated; closed and open channel current levels, red and green dashes; scale bar, 20 pA/50 ms. **(B)** Current–voltage ($i$–V) relationships with pipette solutions at pH 6.0 and 8.5 (left and right panels). Note the increased conductance of 540 pS at pH 8.5 at the cytosolic channel face. **(C)** Open probability vs. $V_p$ plots for these conditions. The underlying data can be found at https://doi.org/10.5281/zenodo.15305314.
(TIF)

**S8 Fig. BVAC is not regulated by cytoplasmic [Ca⁺⁺]. (A)** BVAC recordings with Buffer M in pipette and bath. $V_p$ as indicated; closed and open channel current levels, red and green dashes; scale bar, 10 pA/50 ms. **(B)** Current–voltage ($i$–V) relationship. **(C)** Open probability vs. $V_p$ plot. Solid line, best fit to Eq. 2. The underlying data can be found at https://doi.org/10.5281/zenodo.15305314.
(TIF)

**S9 Fig. Conditional PfCRT knockdown. (A)** Schematic showing CRISPR/Cas9 strategy for the engineered *CRT-KD* line. The *glmS* riboswitch permits conditional regulation of transcript abundance through GlcN-dependent mRNA degradation while DDD allows post-translational regulation through protein denaturation upon TMP removal. **(B)** Ethidium-stained gel showing PCR integration checks with primer positions as indicated in panel (A). Primers are listed in S1 Table. Expected amplicon sizes for each primer pair (in bp): p3-p4, 1856 (WT) and 2422 (*CRT-KD*); p3-p2, 1276 (WT) and 1833 (*CRT-KD*); p1-p4, 1304 (WT) and 1870 (*CRT-KD*). **(C)** Indirect immunofluorescence images of trophozoite-stage *CRT-KD* and WT parasites under indicated conditions and probed with anti-HA antibody. The *CRT-KD* control was grown with TMP and without GlcN. WT was not recognized, indicating antibody specificity. Scale bars, 5 μm.
(TIF)

**S10 Fig. Conditional PfMDR1 knockdown in the Dd2 multidrug-resistant line. (A)** PfMDR1 transmembrane topology showing 12 predicted transmembrane domains and 2 nucleotide-binding domains (NBD) for ATP binding (adapted from ref. [66]). **(B)** CRISPR/Cas9 strategy for the engineered *MDR1-KD* line showing the *glmS* riboswitch. C-terminal epitope tags are followed by a T2A ribosome skip peptide sequence and a neomycin resistance gene (NeoR) to facilitate selection-linked integration. The Southern probe hybridizes to both WT and *MDR1-KD* sequences; restriction sites reflect expected Southern blot cleavage products. **(C)** Gel showing PCR integration checks for indicated parasites and primer pairs (panel (B), S1 Table). Expected amplicon sizes (bp): p5-p6, 756 (WT only); p5-p7, 1291 (*MDR1-KD* only); p5-p9,

1697 (WT) and 2803 (*MDR1-KD*); p8-p9, 1532 (*MDR1-KD* only). (D) Southern blot for *MDR1-KD* clone (*KD*) and WT parent, establishing a single *pfmdr1* copy in the Dd2 wild-type parental line (WT) and complete replacement with the conditional knockdown cassette in *MDR1-KD*. Expected digestion product sizes (bp): BamH1, 3444 (WT), 4591 (*MDR1-KD*), 8935 (plasmid); HindIII, 1700 (WT), 2838 (*MDR1-KD*); NdeI, 2507 (WT), 3654 (*MDR1-KD*), 3660 (plasmid); XbaI, 3607 (WT), 4754 (*MDR1-KD*). Doublets seen in the *MDR1-KD* lanes reflect the genomic and retained plasmid in the transfectant clone (lower and upper bands, respectively). (E) Anti-FLAG immunofluorescence images of *MDR1-KD* grown without and with GlcN, establishing localization at the DV membrane and knockdown. The WT parent is not recognized. Scale bars, 5 μm. (F) Nano-Glo HiBit blot showing PfMDR1 knockdown in *MDR1-KD* with GlcN. The wild-type negative control (WT, left) and aldolase loading control (bottom) are included.
(TIF)

**S11 Fig. PfMDR1 knockdown in 3D7 produces similar phenotypes. (A)** Prevailing model for how *pfmdr1* gene duplication increases transporter abundance and produces resistance to antimalarials including mefloquine. Schematic shows 3D7 with one copy and a mefloquine-resistant line with two copies. Bottom, PfMDR1 is shown on the DV membrane and is color-coded to match gene color in the ribbon. **(B)** Ribbon schematic showing the 3D7 wild-type (WT) and its engineered daughter, $MDR1\text{-}KD_{3D7}$. Modifications and primer positions are indicated. **(C)** PCR integration checks for indicated parasites and primer pairs (panel (B), S1 Table). Expected amplicon sizes (bp): p5-p6, 756 (3D7 only); p5-p7, 1291 ($MDR1\text{-}KD_{3D7}$ only); p5-p9, 1697 (3D7) and 2803 (*MDR1-KD*); p8-p9, 1532 ($MDR1\text{-}KD_{3D7}$ only). **(D)** Anti-FLAG immunoblot showing conditional knockdown of PfMDR1 in $MDR1\text{-}KD_{3D7}$ total cell membranes upon GlcN treatment. The wild-type control and aldolase loading controls are included. **(E)** Mean ± S.E.M. % residual PfMDR1 after knockdown in $MDR1\text{-}KD_{3D7}$ total membranes (cell), quantified using anti-FLAG immunoblots. **(F)** Mean ± S.E.M. expansion of indicated parasites over 5 days. Black circles, control medium; red circles, GlcN pulse; blue circles, chloroquine control (3D7 only). $P = 0.01$ for GlcN pulse in $MDR1\text{-}KD_{3D7}$ but n.s. for 3D7. **(G)** Single channel recordings on $MDR1\text{-}KD_{3D7}$ after PfMDR1 knockdown with GlcN. Buffer A in pipette and bath; $V_p$ as indicated. Red dotted line, closed level. **(H)** % of patches with channels for indicated parasites and GlcN treatment. ns, no statistically significant difference between all tested arms. $P \geq 0.58$ for pairwise comparisons. The underlying data can be found at https://doi.org/10.5281/zenodo.15305314.
(TIF)

**S12 Fig. BVAC is not inhibited by antimalarials, DV transport modulators, or common anion channel blockers.** Single channel recordings on DV from Dd2 parasites with indicated inhibitor present in bath and pipette solutions (Buffer A plus WOS). $V_p$, −60 mV. Red and green dotted lines, closed and open channel levels. These inhibitors do not significantly alter BVAC gating. The underlying data can be found at https://doi.org/10.5281/zenodo.15305314.
(TIF)

**S1 Table. Primers used in this study.**
(XLSX)

**S2 Table. Buffer compositions and additives for patch-clamp in this study.**
(DOCX)

**S1 Video. Organelle-attached patch-clamp of a metabolically active DV.** Hemozoin tumbling is not compromised by patch pipette capture.
(MP4)

**S2 Video. Movie shows a Dd2ᵃᵗᵗᴮ/CRT-GFP-infected erythrocyte.** Live-cell fluorescence imaging shows hemozoin tumbling in the intracellular DV.
(MP4)

**S3 Video. Fluorescence imaging of freshly harvested DV from Dd2^attB/CRT-GFP.**
(MP4)

**S4 Video. Loss of hemozoin tumbling after ~1 h.**
(MP4)

**S1 Raw Images. Uncropped gels and immunoblots used for Figs 3B, 3F, S9B, S10C–S10F, S11C, and S11D.** The raw data for Figs 1D–1F, 2A–2C, 3C, 3D, 3G–3I, 4, S5A–S5F, S6A–S6C, S7A–S7C, S8A–S8C, S11E–S11H, and S12 are available at https://doi.org/10.5281/zenodo.15305314.
(PDF)

## Acknowledgments

We thank M. Fay for help with statistical analysis of single channel abundance per membrane patch in knockdown studies. We thank F. Hoyt and E. Fischer for assistance with transmission electron microscopy studies. We thank D. Jacobus for providing WR99210, BEI Resources for DSM-1, P. Moura for Dd2^attB/CRT-GFP, and T. Wellems for Dd2, HB3, and 3D7 parasite clones. We thank R. Kissinger and A. Mora for creating the Fig 5 illustration.

## Author contributions

**Conceptualization:** Gagandeep S. Saggu, Sanjay A. Desai.

**Formal analysis:** Gagandeep S. Saggu, Jinfeng Shao, Mansoor A. Siddiqui, Maria Traver, Tatiane Macedo-Silva, Joseph Brzostowski, Sanjay A. Desai.

**Investigation:** Gagandeep S. Saggu, Jinfeng Shao, Mansoor A. Siddiqui, Maria Traver, Tatiane Macedo-Silva, Joseph Brzostowski.

**Methodology:** Gagandeep S. Saggu, Jinfeng Shao, Maria Traver, Tatiane Macedo-Silva, Joseph Brzostowski, Sanjay A. Desai.

**Project administration:** Sanjay A. Desai.

**Software:** Sanjay A. Desai.

**Writing – original draft:** Gagandeep S. Saggu, Sanjay A. Desai.

**Writing – review & editing:** Gagandeep S. Saggu, Jinfeng Shao, Mansoor A. Siddiqui, Maria Traver, Tatiane Macedo-Silva, Joseph Brzostowski, Sanjay A. Desai.

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
