## [Editor Report · Decision Letter 0]

12 Dec 2024

Dear Dr Desai,

Thank you for submitting your manuscript entitled "A large anion channel for malaria parasite digestive vacuole acidification and amino acid export" for consideration as a Research Article by PLOS Biology.

Your manuscript has now been evaluated by the PLOS Biology editorial staff, as well as by an academic editor with relevant expertise, and I am writing to let you know that we would like to send your submission out for external peer review.

Once your full submission is complete, your paper will undergo a series of checks in preparation for peer review. After your manuscript has passed the checks it will be sent out for review. To provide the metadata for your submission, please Login to Editorial Manager (https://www.editorialmanager.com/pbiology) within two working days, i.e. by Dec 14 2024 11:59PM.

Kind regards,

Melissa

Melissa Vazquez Hernandez, Ph.D.

Associate Editor

PLOS Biology

---

## [Decision Letter · Decision Letter 1]

31 Jan 2025

Dear Dr Desai,

Thank you for your patience while your manuscript "A large anion channel for malaria parasite digestive vacuole acidification and amino acid export " was peer-reviewed at PLOS Biology. It has now been evaluated by the PLOS Biology editors, an Academic Editor with relevant expertise, and by two independent reviewers.

In light of the reviews, which you will find at the end of this email, we would like to invite you to revise the work to thoroughly address the reviewers' reports. As you will see below, the reviewers are positive about the relevance and novelty of the study, yet some concerns have raised during revision. Reviewer 1 requires some clarifications and suggests to evaluate the integrity of the digestive vacuoles, and to suggest potential protein candidates involved in the anion conductance. Reivewer 2 has a major concern regarding the recording buffer and asks the authors to prove channel activity at low pH. We agree with all reviewer concerns and would require some additional experimental revisions to address them, as well as discussion about the potential protein candidates and transparency in the experiments, as we consider that this would strengthen the work.

Given the extent of revision needed, we cannot make a decision about publication until we have seen the revised manuscript and your response to the reviewers' comments. Your revised manuscript is likely to be sent for further evaluation by all or a subset of the reviewers.

**IMPORTANT - SUBMITTING YOUR REVISION**

*Re-submission Checklist*

*Published Peer Review*

*PLOS Data Policy*

*Blot and Gel Data Policy*

Sincerely,

Melissa

Melissa Vazquez Hernandez, Ph.D.

Associate Editor

PLOS Biology

REVIEWERS' COMMENTS:

Reviewer #1:

The manuscript describes the discovery of an anion conductance in the digestive vacuolar membrane of Plasmodium falciparum, the human malaria parasite. To achieve this, the authors developed an improved method for isolating intact digestive vacuoles, which presents significant challenges due to the sharp heme crystals within this organelle that can compromise membrane integrity. Additionally, the authors constructed particularly small patch-clamp pipettes, allowing them to successfully access the tiny food vacuoles for electrophysiological studies. These technical innovations are commendable and critical for the success of the study.

After overcoming these methodological obstacles, the authors characterized the anion conductance, demonstrating its voltage-dependent permeability to various inorganic and organic anions. Importantly, they utilized genetically engineered knockdown mutants to show that the anion conductance operates independently of PfCRT and PfMDR1, two well-characterized transporters of the digestive vacuolar membrane. Furthermore, the authors investigated the druggability of the anion conductance, employing antimalarial drugs and specific anion channel inhibitors. The study concludes with a proposed model suggesting that this anion conductance plays a key role in maintaining electroneutrality in the digestive vacuole, which is essential for managing the acidic environment within this organelle.

Overall, this is a well-written, topical, and compelling study that significantly advances our understanding of the biology of the digestive vacuole and its role in P. falciparum. However, several aspects require clarification or further consideration:

Queries and Suggestions for Improvement

1. Integrity of the Digestive Vacuoles

The experiments rely on the assumption that the digestive vacuolar membrane remains intact during isolation. While the authors use the movement of hemozoin crystals as an indicator of vacuolar integrity, this should be substantiated with a more robust assay. For example, the authors could measure the uptake of tritiated water and [¹⁴C] or tritiated inulin to determine the membrane's integrity quantitatively. Such an assay would strengthen confidence in the physiological relevance of their findings.

2. Relevant References

o To provide a more comprehensive context for their results, the authors should cite article PMID: 35867395, which describes a conditional PfCRT knockdown mutant and its physiological implications.

o In discussing PfMDR1 and its role in drug resistance, reference to https://doi.org/10.1371/journal.pbio.3001616 is necessary.

o Additionally, when addressing digestive vacuolar acidification, the authors should include PMID: 37463201 in their references. These citations would strengthen the manuscript's connection to relevant prior work.

3. Molecular Identity of the Anion Conductance

While the study robustly demonstrates the existence of an anion conductance through electrophysiological evidence, its molecular identity remains unknown. The reviewer recognizes that full genetic characterization is beyond the scope of this work. However, given the existence of proteomic studies on the digestive vacuole (e.g., PMID: 21136929 and PMID: 28784333), it would be valuable for the authors to suggest potential candidates or provide a discussion of likely proteins involved. This would enrich the manuscript and offer a starting point for future investigations.

By addressing these points, the manuscript would provide an even stronger foundation for understanding the role of anion conductance in P. falciparum biology and its potential as a target for antimalarial drug development.

Reviewer #2:

Saggu et al. "A large anion channel for malaria parasite digestive vacuole acidification and amino acid transport" establishes of a chloride channel in the malaria parasite digestive vacuole (DV) membrane. With great skill the authors perform patch clamp on the vacuole to characterize channel activity on the organelle's membrane. They go on to show voltage-clamp current recordings consistent with a single channel type. By varying the ion composition of the bath and pipette buffers the authors establish that the channel preferentially conducts anions up to the size of an amino acid. The authors go on to demonstrate that the channel is found in at least 3 commonly used strains of P. falciparum and that known transporters at the DV are not the cause of the channel activity, the channel is also insensitive to a range of known anti-malarials. While the channel properties are consistent with those of anion channels of other organisms, known channel inhibitors have no significant effect on the channel's activity.

The paper is a significant contribution presenting a to the community of malaria and ion channel research, adding high resolution functional (electrophysiology) data that explains Plasmodium spp. digestive vacuole physiology. The electrophysiological characterization of a novel organellar channel is a great achievement. I recommend the work to be published and made accessible to the community.

I have one larger and few smaller concerns that I hope can help with work and that the authors may want to address.

My one major concern is about the recording buffer. In the methods Na-K buffer is mentioned to contain 70 mM KCl, 70 mM NaCl, 2 mM CaCl2, 2.5 mM MgCl2, 10 mM HEPES, 10 mM MES, pH 7.4, for modifications only major salts are mentioned. It is unclear if Ca and Mg are always present. For channel activity most relevant and interesting would likely be pH and Ca++ (see for example Riederer et al 2023). As there is a large pore in the DV membrane it appears the DV lumen would be clamped to the bath pH and [Ca++]. pH sets the protonation of the protein domains in the lumen which is relevant for proper protein activity, while pH and Calcium appear most relevant for regulation of channels in lysosomal compartments. I would expect the phycological pH of the DV lumen to be ~5 and Ca++ ~0.5mM. It would be helpful to see the physiology of the recording conditions discussed. If the researchers can add recordings, it would be most valuable to probe channel activity at low pH, while it can be interesting to also alter Ca++.

Further concerns in order as they appear in the text:

I don't think the mechanism for hemozoin tumbling is known and should maybe not be promoted as indication of healthy DV function until then. My personal impression is that the movement depends strongly on light exposure.

The method to harvest DVs without detergent seems missing.

Exact buffer compositions (as mentioned above) and temperatures at which the experiments where performed seem not given but are necessary to understand the data.

The calculations for the reversal potentials and permeability ratios should be given. Assuming room temperature for the experiments, I calculated P_Cl/P_cat = 2.2 a little lower than what is reported.

I am somewhat surprised that no chloride was present in the pipette determining P_Cl/P_cat. The access resistance must have been very high without charge carriers, and a AgCl electrode would also produce an indeterminate voltage offset without use of a salt bridge in absence of chloride. Is sorbitol the only solute in the pipette, no other salts are present? It would perhaps be most unambiguous to provide all electrophysiology buffer conditions in a supplementary table.

Glutamate is a relatively large ion that may reduce buffer conductivity. It may be that the lower glutamate pore conductance scales with the buffer conductivity and can be rationalized in that way.

It would be helpful to indicate 0 pA levels on the current traces in the figures to estimate if the channel fully closes or remains partially open. It would be helpful if the authors can comment on full/partial closure as well. Alternatively, some raw current traces can by nice as supplement.

Supplementary figures S2A, S2B, S6G and S7 are difficult to interpret without quantification. If there is enough data for i/i_max vs V (p_open vs V), single channel conductance and dwell times it would be good to show. Especially for the inhibitors, in Fig S7 it is difficult to make out that they are truly the same. To my eye it seems that with DIDS and phloridzin the channel flickers faster?

---

## [Decision Letter · Decision Letter 2]

24 Apr 2025

Dear Dr Desai,

Thank you for your patience while we considered your revised manuscript "A large anion channel for malaria parasite digestive vacuole acidification and amino acid export" for publication as a Research Article at PLOS Biology. This revised version of your manuscript has been evaluated by the PLOS Biology editors, the Academic Editor and two of the original reviewers.

Based on the reviews, we are likely to accept this manuscript for publication, provided you satisfactorily address the remaining points raised by the reviewer 2. In this case, while we do not expect new experiments, adjusting the claims considering the reviewer's points is mandatory for publication. Please also make sure to address the following data and other policy-related requests.

a) We routinely suggest changes to titles to ensure maximum accessibility for a broad, non-specialist readership, and to ensure they reflect the contents of the paper. In this case, we would suggest a minor edit to the title, as follows. Please ensure you change both the manuscript file and the online submission system, as they need to match for final acceptance:

"Identification of a large anion channel required for digestive vacuole acidification and amino acid export in Plasmodium falciparum"

Please supply the numerical values either in the a supplementary file or as a permanent DOI’d deposition for the following figures:

Figure 1DEF, 2ABC, 3CDGHI, 4, S3ABCD, S4, S5A-F, S6ABC, S7ABC, S8ABC, S11EFGH, S12

c) Please cite the location of the data clearly in all relevant main and supplementary Figure legends, e.g. “The data underlying this Figure can be found in S1 Data” or “The data underlying this Figure can be found in https://doi.org/10.5281/zenodo.XXXXX”

d) We require the original, uncropped and minimally adjusted images supporting all blot and gel results reported in the Figures 3BF, S9B, S10CDF, S11CD

We will require these files before a manuscript can be accepted so please prepare and upload them now. Please carefully read our guidelines for how to prepare and upload this data: https://journals.plos.org/plosbiology/s/figures#loc-blot-and-gel-reporting-requirements

e) Please ensure that your Data Statement in the submission system accurately describes where your data can be found and is in final format, as it will be published as written there.

*Many thanks for providing the underlying code in GitHub. However, because Github depositions can be readily changed or deleted, please make a permanent DOI’d copy (e.g. in Zenodo) and provide this URL in the manuscript and Data Availability Statement.

f) Per journal policy, if you have generated any custom code during the course of this investigation, please make it available without restrictions upon publication. Please ensure that the code is sufficiently well documented and reusable, and that your Data Statement in the Editorial Manager submission system accurately describes where your code can be found.

e) Please note that per journal policy, the model system/species (Plasmodium falciparum) studied should be clearly stated in the abstract of your manuscript.

g) Thank you for providing the funding institutions, but we will also need the grant numbers.

We expect to receive your revised manuscript within two weeks.

*Published Peer Review History*

*Press*

Sincerely,

Melissa

Melissa Vazquez Hernandez, Ph.D.

Associate Editor

PLOS Biology

REVIEWERS' COMMENTS:

Reviewer #1: The reviewer thanks the authors for the constructive scientific dialogue. The authors have address my comments in full. No further suggestions.

Reviewer #2:

The authors thoroughly worked on the comments, and it is great to see the some of the questions addressed with new data. I appreciate the reporting of the experimental recording conditions and the availability of current traces.

Naturally I would have addressed some issues differently. While I appreciate the attempt to control pH I will argue that the results are not interpretable with the available data. The lack of pH control/quantification introduce a new major weakness that could be addressed either by weakening the conclusions or new data.

It is interesting that the E(1)GFP doesn't seem to respond to pH. I didn't find the protocol described to calibrate pH so I can't evaluate the technical side. I would think that it is unusual that the fluorescence is intact while the pH sensitivity is lost, rather I'd expect the fluorescence to be lost entirely. It is unfortunate the pH sensor isn't working as one could envision many experiments to explore pH homeostasis of the DV with a working pH sensor. In any case, it could be worthwhile switching to a different pH sensor for the reasons outlined below.

I agree with the authors response to reviewer 1's comment on the intactness of the DV. Since all recordings are performed in an attached configuration, it doesn't matter if the membrane, as a whole, is intact (as long as a patch can seal and the channels in the patch can reasonably expected to be folded well). On the flip side, I don't quite understand why the authors insist that the ion concentration of the DV interior would be unaffected by the bath ions. The channel never seems to fully close and is able to permeate all tested ions. Some of the recordings were done in presence of ATP, GTP, phosphocreatine which is theoretically helping to maintain low pH in the DV but without ATP (or the other energy source) I don't see how the DV would hold any ionic gradient.

A working pH sensor would help understand any potential pH dependent gating mechanisms in a well controlled manner. I am not sure the right papers are cited to argue for intact pH homeostasis of the isolated vacuole and the use of the employed compounds. I think PMID 37463201 measures cell proliferation in response to concanamycin and PMID 3905824 looks at digitonin released parasites but the DV is not isolated. NH4Cl only has an effect if the DV is acidic to begin with, similar for concanamycin. To my understanding, for the purpose of characterizing BVAC dependence on vacuolar pH, the authors are lacking a working pH sensor and can't assess the effect of the compounds used to alter the pH. Hence, I'd suggest to not interpret the results that rely on the presumed change of luminal pH.

Two minor points I want to ask to consider:

Ionic mobility (somewhat correlated to ion size) has an effect on conductivity. This can be used to scale conductivity for larger ions but also to correct for junction potentials. For general understanding somewhat relevant maybe is PMID: 7715244. I would love to talk to the authors about it. I feel we have some misunderstanding of each others' comments.

For the 0 current indication, I understand that there will be leak currents. However, it would be informative to see if the channel fully closes. If there was a leak current proportional to the number of channels in the patch, one could assume this is due to only incomplete closure of the channel. Perhaps the right thing to ask for would be raw recordings from a voltage step protocol.

---

## [Editor Report · Decision Letter 3]

8 May 2025

Dear Sanjay,

Thank you for the submission of your revised Research Article "Identification of a large anion channel required for digestive vacuole acidification and amino acid export in Plasmodium falciparum" for publication in PLOS Biology. On behalf of my colleagues and the Academic Editor, Tania F. de Koning-Ward, I am pleased to say that we can in principle accept your manuscript for publication, provided you address any remaining formatting and reporting issues. These will be detailed in an email you should receive within 2-3 business days from our colleagues in the journal operations team; no action is required from you until then. Please note that we will not be able to formally accept your manuscript and schedule it for publication until you have completed any requested changes.

IMPORTANT: I've asked my colleagues to include the following editorial request alongside their own: I note that in your data availability statement you say, Additional code tailored specifically for the channel described here is available upon request." We do not allow on-request statements, so please include this additional custom code in Zenodo, and change the data availability statement accordingly.

PRESS

Sincerely,

Melissa

Melissa Vazquez Hernandez, Ph.D., Ph.D.

Associate Editor

PLOS Biology
